# TOWARD OPEN-ENDED EMBODIED TASKS SOLVING

## ABSTRACT

Empowering embodied agents, such as robots, with Artificial Intelligence (AI) has become increasingly important in recent years. A major challenge is task open-endedness. In practice, robots often need to perform tasks with novel goals that are multifaceted, dynamic, lack a definitive "end-state", and were not encountered during training. To tackle this problem, this paper introduces *Diffusion for Open-ended Goals* (DOG), a novel framework designed to enable embodied AI to plan and act flexibly and dynamically for open-ended task goals. DOG synergizes the generative prowess of diffusion models with state-of-the-art, training-free guidance techniques to adaptively perform online planning and control. Our evaluations demonstrate that DOG can handle various kinds of novel task goals not seen during training, in both maze navigation and robot control problems. Our work sheds light on enhancing embodied AI's adaptability and competency in tackling open-ended goals.

## 1 INTRODUCTION

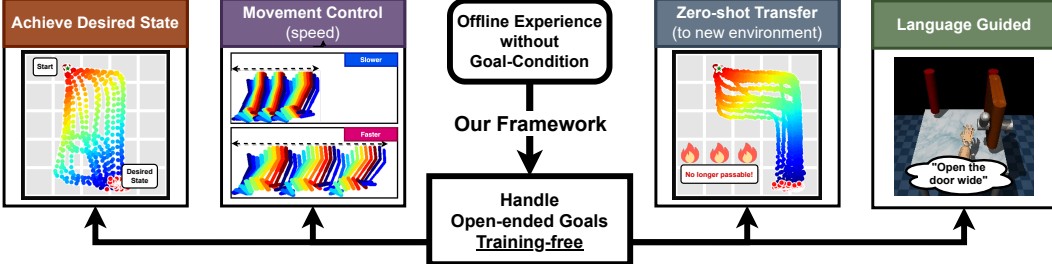

Figure 1: **Handling open-ended goals.**

Task solving for **open-ended goals** (Fig. 1) in embodied artificial intelligence (AI) (Jin & Zhang, 2020) represent a cornerstone in the pursuit of creating machines that can assist humans in real-world (Taylor et al., 2016). Unlike traditional AI that operates in virtual realms or specific, constrained settings (Silver et al., 2016), embodied AI is situated in the physical world—think robots, drones, or self-driving cars. Here, the utility is not merely in solving a specific problem but in the system's ability to perform a broad range of tasks, enabling everything from advanced automation to assistive technologies for the disabled, much like humans and animals do.

Yet, this endeavor presents a myriad of challenges. Real-world tasks with open-ended goals are highly diverse and often cannot be described by a single variable or a single type of variables. For example, an embodied agent tasked with "assisting in household chores" would require the capabilities to perform various tasks, from vacuuming to cooking, while adapting to new challenges and human preferences over time. These goals are almost impossible to fully cover in learning. The inherent complexity and variability of such goals necessitate a significant advancement in decision-making capacity.

To create embodied AI that can flexibly handle open-ended goals, both the knowledge about the world and skills of motor actions need to be equipped. Only until recent times, a handful of works (Driess et al., 2023; Dai et al., 2023) started the attempts for ambition by leveraging the real-world knowledge from pre-trained vision (Rombach et al., 2022) and/or language (Brown et al., 2020)

foundation models. On the other hand, Stooke et al. (2021); Bauer et al. (2023) endeavors to perform large-scale multi-task training in a game world so that the agent can quickly adapt to novel tasks. These works are worthy of recognition on the path to embodied AI that can truly tackle open-ended tasks. Nonetheless, these studies are still trapped by the conventional *goal-as-an-input* paradigm (Fig. 2 Left), and thus the flexibility of goals is limited (e.g., if a robot is trained to go anywhere in the world, after training it cannot be asked to keep away from somewhere.).

In the presence of these challenges, we propose a novel framework for solving embodied planning and control for open-ended goals. This work is a trial to approach the ultimate embodied AI that can assist people with diverse tasks such as healthcare, driving, and housework, though further endeavor is, of course, still largely demanded. Here, we empower embodied AI with the recent advance of diffusion models (Ho et al., 2020) and training-free guidance (Yu et al., 2023b) to overcome the challenges of open-ended goals. We refer to our framework as *Diffusion for Open-ended Goals* **(DOG)**. Our contributions can be summarized as follows:

1. By borrowing the concept of energy functions into the Markov decision process, we provide a novel formulation for modeling open-ended embodied tasks. This scheme enjoys much higher flexibility than traditional goal-conditioned decision-making methods.

2. We propose the DOG framework to solve open-ended tasks. In the training phase, the unconditional diffusion models are employed to learn world knowledge from the offline experience without goal-conditioning. During the inference stage, the agent would make plans and act based on the world knowledge in diffusion models and the knowledge of goals by performing energy minimization.

3. We evaluate the proposed method in a wide range of embodied decision-making tasks including maze navigation, robot movement control, and robot arm manipulation. DOG is shown to effectively and flexibly handle diverse goals that are not involved in training.

## 2 PRELIMINARIES

### 2.1 MARKOV DECISION PROCESS

A *Markov Decision Process* (MDP) (Bellman, 1957) is a mathematical framework used for modeling decision-making problems. An MDP is formally defined as a tuple $(\mathcal{S}, \mathcal{A}, \mathcal{P}, \mathcal{R})$, where $\mathcal{S}$ is a space of states, $\mathcal{A}$ is a space of actions, $\mathcal{P} : \mathcal{S} \times \mathcal{S} \to \mathcal{A}$ is the state transition probability function, and $\mathcal{R}$ is the reward function. Solving an MDP involves finding a policy $\pi : \mathcal{S} \to \mathcal{A}$, which is a mapping from states to actions, that maximizes the expected sum of rewards over time. Our work borrows the terminologies and notations from MDPs, while we consider general task goals (Sec. 3) instead of getting more rewards as in original MDPs.

### 2.2 DIFFUSION MODELS AND CLASSIFIER GUIDANCE

Diffusion models have emerged as a powerful framework for generative modeling in deep learning (Ho et al., 2020). These models iteratively refine a noisy initial input towards a data sample through a series of reverse-diffusion steps. Diffusion models are a popular type of diffusion, whose core idea is to estimate the score function $\nabla_{\Omega^n} \log p(\Omega^n)$, where $\Omega^n$ is the noisy data at the time step $t$. Given a random noise $\Omega^n$, diffusion models progressively predict $\Omega^{n-1}$ from $\Omega^n$ using the estimated score $\Omega^{n-1} = (1 + \frac{1}{2}\beta^n)\Omega^n + \beta^n \nabla_{\Omega^n} \log p(\Omega^n) + \sqrt{\beta^n}\epsilon, \quad n \leq N$, where $\beta^n$ is a coefficient and $\epsilon \sim \mathcal{N}(0, I)$ is Gaussian noise. During the training process, the goal is to learn a neural network $s_\theta(x^n, t) \approx \nabla_{x^n} \log p(x^n)$, which will be used to replace $\nabla_{\Omega^n} \log p(\Omega^n)$ during inference.

A unique advantage of diffusion models is **training-free** conditional generation. To allow conditional guidance, one should compute $\nabla_{\Omega^n} \log p(\Omega^n | c)$, where $c$ is the condition. Using the Bayesian formula, the conditional score function can be written as two terms: $\nabla_{\Omega^n} \log p(\Omega^n | c) = \nabla_{\Omega^n} \log p(\Omega^n) + \nabla_{\Omega^n} \log p(c | \Omega^n)$, where $\nabla_{\Omega^n} \log p(\Omega^n)$ is the unconditional score obtained by the pretrained diffusion model and $p(c | \Omega^n) \propto \exp(-\ell(x^n))$, where $\ell(\cdot)$ is an easy-to-compute loss function. For example, in image generation, unconditional diffusion can generate natural images, and $p(c | \Omega^n)$ is a classifier. By adding the gradient of the classifier to the pretrained diffusion neural network, the model can perform conditional generation based on classes.

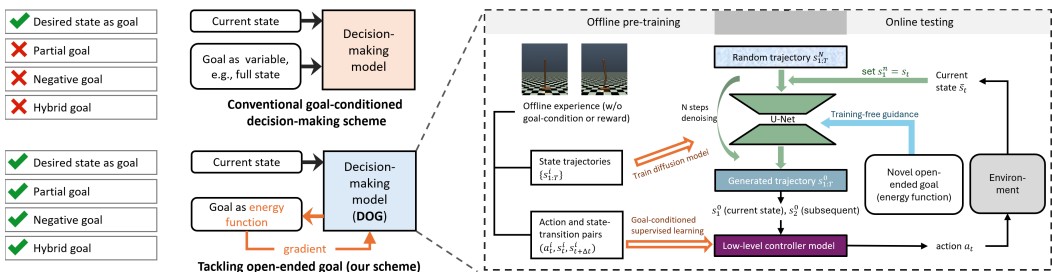

Figure 2: Difference from conventional goal-conditioned decision-making (left) and diagram of our framework (right).

# 3 OPEN-ENDED TASK GOALS FOR EMBODIED AI

Our work aims to solve practical tasks with embodied agents (Jin & Zhang, 2020; Gupta et al., 2021) such as robots. Embodied decision-making faces distinguished challenges from computer vision (Rombach et al., 2022) or nature language processing (Brown et al., 2020). First, embodied tasks are constrained by real-world physics. Generative models such as diffusion models for image generation can "draw" the pixels with any color, whereas a real-world painter can only use the colors available in the palette. Second, decision-making involves sequence modeling, which requires an agent to predict and plan actions over time. This is different from traditional machine learning tasks, where the focus is on finding a single optimal output for a given input. Oftentimes we care about multiple steps rather than a single state, e.g, asking a robot to move in circles. Therefore, tackling open-ended task goals in embodied AI becomes challenging. To meet the need of describing open-ended goals, we define an *open-ended goal*, using the notations in MDP, in an environment[1] with the format of energy function of a contiguous sequence of states (suppose the task horizon is $T$, which can be finite or infinite):

$$\text{To minimize} \qquad g(s_{1:T}) : \mathbb{R}^{T \cdot n_s} \to \mathbb{R}, \tag{1}$$

where state $s_t \in \mathcal{S}$ and $n_s$ is its dimension (Here we consider continuous state and actions for embodied decision-making problems while the ideas can also apply to discrete actions.) The *goal energy function* $g$ is any differentiable [2] function with a real scalar output. Note that $g$ is not fixed nor pre-determined before training, but can be any function when needed. To consider an intuitive example, imagine that a nanny robot is first trained to understand the structure of your house, then you may ask the robot to perform various tasks in the house as long as you can define a goal function, e.g., using CLIP (Radford et al., 2021) to evaluate how much the robot's visual observation is consistent with the embedding of "clean, tidy". This way of defining a goal is fundamentally different from the conventional way of treating the goal as a variable (so it can be an input argument to the decision-making model) (Andrychowicz et al., 2017; Liu et al., 2022).

Our goal definition by energy function (Eq. 1) offers several notable advantages (Fig. 2). Some intuitive examples are explained as follows. Suppose a robot's state at step $t$ is represented by horizontal and vertical coordinates $(x_t, y_t)$. The goal can be to

- **partial goal:** Go left (smaller $x$) regardless of $y$ by letting $c(x_{1:T}, y_{1:T}) = \sum_{1:T} x_t$;
- **negative goal:** Avoid a position $(\hat{x}, \hat{y})$ via $c(x_{1:T}, y_{1:T}) = -\sum_{1:T}((x_t - \hat{x})^2 + (y_t - \hat{y})^2)$;
- **sequence goal:** Moving close to a given trajectory $(\hat{x}_{1:T}, \hat{y}_{1:T})$ with $c(x_{1:T}, y_{1:T}) = \sum_{1:T}((x_t - \hat{x}_t)^2 + (y_t - \hat{y}_t)^2)$;
- **hybrid goal:** The combination of several goal energy functions by summation.

Due to the diverse input types, these goals cannot be directly handled by *goal-as-input* scheme (Fig. 2) without training.

---

[1]We consider an environment as a world with fixed physical laws, such as the earth, and various tasks (goals) can be defined in this environment.

[2]In this work, we consider a differentiable goal energy function since it can already be applied to many cases. For surrogate gradient or Monte-Carlo method could be used, which remains as future work.

## 4 METHODS

### 4.1 OBJECTIVES AND CHALLENGES

In this section, we define the main goals and inherent challenges of our proposed method, formulated for a two-phase approach. The primary objectives of the training and testing stages are as follows:

- **Training Stage.** The agent is designed to learn and internalize knowledge of the environment from offline data. The testing goal is unknown.

- **Testing Stage.** Given a novel task, the agent must interpret of goal description $g \in \mathcal{G}$ and generate and then execute plans in alignment with the knowledge obtained during training to complete the goal $g$.

The realization of these objectives poses significant challenges within the existing frameworks. **Traditional offline RL algorithms** are often black-box optimization processes, usually targeting singular goals and exhibiting resilience to modifications for diverse objectives. **Goal-conditioned algorithms** can only work when the task is achieve specific states $s$ (refer to A.3 for more discussion). **Large language models** (Huang et al., 2023) can handle open-goal tasks but struggle to utilize offline data to learn the environment transition. In summary, a gap exists for methods that can both leverage offline data for environmental learning during training and can accommodate diverse goals during testing.

### 4.2 TRAINING STAGE: UNDERSTANDING WORLD DYNAMICS BY DIFFUSION MODEL

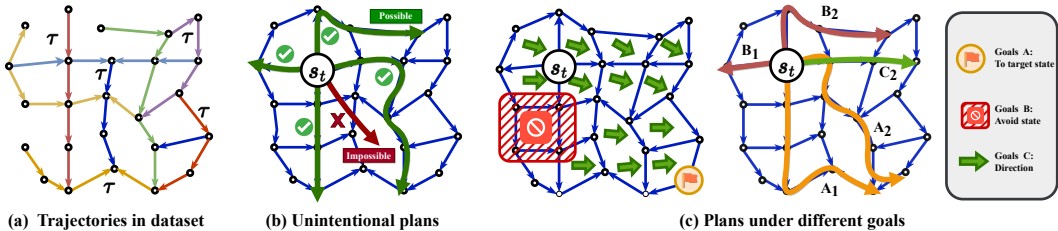

(a) Trajectories in dataset     (b) Unintentional plans     (c) Plans under different goals

Figure 3: **Illustration of the proposed framework**.

Given an offline dataset $D = \{\tau_i\}_i^{N_d}$ consisting of $N_d$ trajectories $\{\tau_i\}_i^{N_d}$ as depicted in Fig. 3a, where each line of identical color represents a distinct trajectory, our objective is to develop a parameterized probabilistic model, denoted as $p_\theta(\Omega|s_t)$. In this model, $\Omega = \{s_{t+i}\}_{i=1}^T$ signifies the observation sequence of the succeeding $T$ steps from $s_t$, intending to emulate the data distribution $p(\Omega|s_t)$ found in the offline dataset. Possessing such a model enables the prediction of plausible future trajectories originating from $s_t$, as illustrated in Fig. 3b. The training can be finished by minimizing the following:

$$\ell_\theta(\cdot) = \mathbb{E}_{(s,\Omega)\sim D}||p_\theta(\cdot|s), \Omega||_2 \tag{2}$$

### 4.3 TESTING STAGE: OPEN-ENDED GOAL-CONDITIONED PLANNING

Merely simulating the data distribution $p(\Omega|s_t)$ is inadequate for planning with open-ended goals. The generated $\Omega^* \sim p_\theta(\Omega|s_t)$ largely mimics the behavior of the dataset, making it suboptimal or unavailable for innovative applications since it only replicates existing patterns. For more utility, the sampling process needs to be aligned with the goal condition $g$. However, accomplishing this is challenging, given that the available resources are confined to a trained model $p_\theta$ and a goal-descriptive function $g$.

To mitigate the identified limitations, we integrate innovations from conditional generative strategies, widely acknowledged in the field of computer vision, into our planning process. Specifically, we utilize the classifier guided diffusion model (Dhariwal & Nichol, 2021; Yu et al., 2023b), recognized for its expertise in creating sophisticated content. This model facilitates the generation of optimal trajectories that are conditioned on precise goals, eliminating the need for supplemental

training. By doing so, it expedites the sampling process and ensures adherence to defined objectives, proficiently fulfilling our requirements and presenting a viable resolution to our challenges.

The training phase proceeds conventionally to train a U-Net (Ronneberger et al., 2015) to learn the score function $s_\theta(\Omega^n, n) \approx \nabla_{\Omega^n} \log p_t(\Omega^n)$. During the test phase, we define the conditional score function as $\exp(-\eta g(\cdot))$ and calculate the gradient

$$\mathbf{grad}^n = \nabla_{\Omega^n} \log p_t(c|\Omega^n) \approx \nabla_{\Omega^n} \log \exp(-\eta g(\Omega^0)) = -\eta \nabla_{\Omega^n} g(\Omega^0)$$

where $c$ represents the goal, $\bar{\Omega}^0 = \sqrt{\bar{\alpha}^n}\Omega^n + \sqrt{1 - \bar{\alpha}^n}\epsilon$ is a denoised version of $\Omega^n$, $g$ is the goal energy function defined in equation 1, and the approximation follows (Chung et al., 2022). Then the reversed diffusion process becomes:

$$\Omega^{n-1} = (1 + \frac{1}{2}\beta^n)\Omega^n + \beta_t s_\theta(\Omega^n, n) + \sqrt{\beta^n}\epsilon + \mathbf{grad}^n, \quad n \leq N \tag{3}$$

Note that, depending on whether our goal needs to include the history, we may concatate the history into the decision process to guide the generation of future plans.

By implementing this methodology, we facilitate efficient generation conforming to the goal condition $g$. This ensures that the generated trajectories are not only optimal but are also in alignment with predetermined objectives, enhancing the model's versatility and reliability across various applications.

## 4.4 PLAN EXECUTING

Once the plans are generated, an actor is required to execute them. Our framework is designed to be flexible, allowing the incorporation of various types of actors, provided they are capable of directing the agent according to the intended state transitions. Several optional can be made for the plan executor which selects action $a_t$ at $s_t$ to achieve the middle waypoint $\hat{s} \in \Omega$. We summarize the possible implementation in Sec. F.

Here, we elaborate on a supervised learning implementation, drawing ideas from Hindsight Experience Replay (HER) (Andrychowicz et al., 2017). The executor is a mapping function $p_\phi : \mathcal{S} \times \mathcal{S} \to \mathcal{A}$. Training is conducted by sampling state pairs $s, a, s'$, where $s$ and $s'$ are within one episode with an interval of $t \sim \text{Uniform}(1, ..., T_a)$. In this context, $T_a$ represents the maximum tolerance of state shifting, allowing us to sample non-adjacent states for a multi-step target state $\hat{s}$ from the planner, avoiding the constraints of only sampling adjacent states. The recorded action at state $s$ serves as the predictive label. This approach enables utilization of offline data to deduce the action needed to reach $\hat{s}$ from $s$. The training loss is expressed as:

$$\ell_\phi(\cdot) = \mathbb{E}_{(s,a,\hat{s})\sim D} \|p_\phi(a|s, \hat{s})\|_2 \tag{4}$$

Since our executor to multiple-step actions, the generated plan can be preserved and deployed over multiple steps. A summary of the algorithm is provided in Alg. 1 in Sec. E

## 5 RESULTS

### 5.1 EXPERIMENT SETTINGS AND IMPLEMENTATION DETAILS

#### 5.1.1 IMPLEMENTATION DETAILS

We defer the model structure design and parameter setting to the Appendix. We summarize the details of the design goal function in B.2. We defer environment introduction to C.

### 5.2 MAZE NAVIGATION IN OPEN-ENDED SCENARIOS

Our methodology is adept at generating plausible plan distributions for future $H$ steps. As depicted in Fig. 4a, when the agent performs rollouts without any guidance, it is observable that the endpoints are distributed randomly across the map. This random distribution represents all the possible future distributions of the agent interacting with the environment. The source model is trained using 1 million steps of offline data, with the generating policy consistent with D4RL (Fu et al., 2020).

Nevertheless, the utility of this method is maximized when it is conditioned on a specific goal as mentioned previously. Here, we illustrate various meaningful goal types within 2D Maze environments to showcase the extensive applicability of our methods. The original maze is devoid of obstacles, enabling an unobstructed view of the episode distribution. For each setting, we present 100 episodes both from the same start point (marked as green star). The end point of the trajectories are marked are red point. For clear visualization, we only show 10 full trajectories, each as one line.

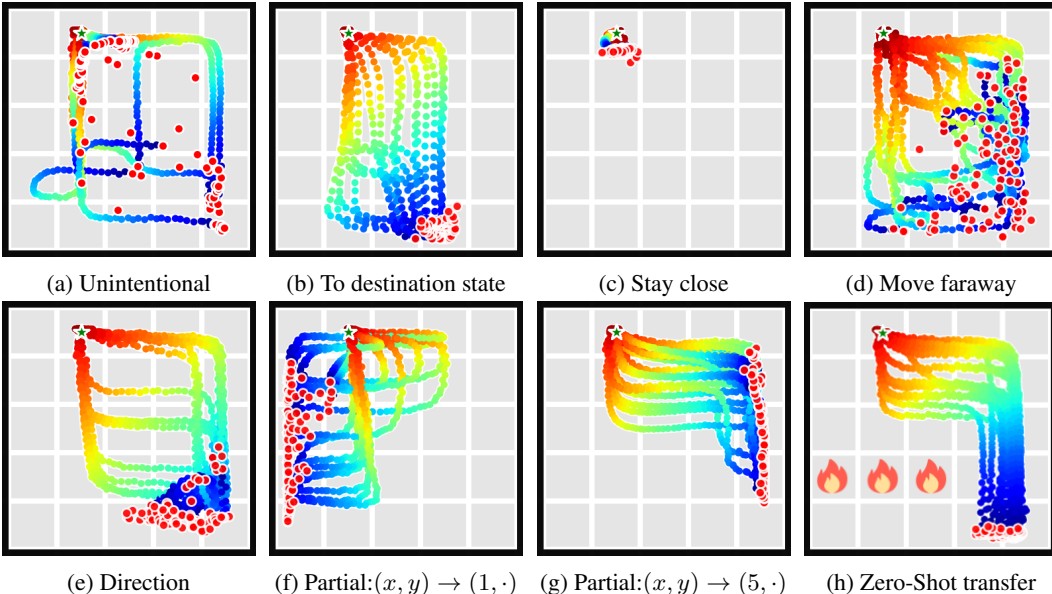

Figure 4: Exploring Open-Ended Goals in Maze2D Environments.

**Goal as State.** In Fig. 4b, we exhibit our proficiency in addressing traditional goal-conditioned RL. The agent is prompted to navigate to a designated location, it shows that it explores varied trajectories to reach the location.

**Partial goal.** Often, our objectives align only partially with observed states, requiring more flexibility in goal setting. Consider a scenario where the goal is to stand higher, however, typically requires the specification of all observation $s$ values. In Fig.4f and Fig.4f, we instruct the agent to navigate to locations where the $x$ value is either 5 or 1, without specifying the $y$ value. This allows the agent to identify viable end $(x, y)$ pairs and the corresponding paths, demonstrating our method's adaptability to partial goals.

**Relative goal.** Goal can be the relationship between states. In Fig. 4c- 4d, our goal is to control the moving distance, which could be calculated as the sum of the distance of consecutive points. Corresponding plans would be generated.

**Non-specific Goal.** Often, our objective isn't state-specific but aims at maximizing or minimizing some certain property, like speed or height. Traditional goal-conditioned RL falters in such rudimentary and prevalent cases. In Fig. 4e, we direct the agent towards a direction (right-bottom), resulting in episode ends congregating around the right-bottom corner, substantiating our aptitude in managing directional goals.

**Zero-shot transfer to new environment.** It's commonplace to deploy agents in environments distinct from their training grounds. Humans, leveraging knowledge acquired during training, can adapt when informed of changes; contrastingly, conventional RL frameworks often falter in integrating new insights into policies. Fig. 4h exemplifies our algorithm's prowess in zero-shot adaptability. When deployed in a maze with impassable grids, the agent is directed to attain the goal without traversing these zones and successfully formulates plans in compliance with these stipulations.

**Hybrid goals** Our algorithm transcends mere single-goal accomplishments. By incorporating the gradients of multiple goals, the agent endeavors to address problems under varied conditions. Fig. 4h

displays outcomes where the agent concurrently avoids fire and attains the goal position. Fig. 4e guide the $x$ and $y$ position at the same time.

## 5.3 MuJoCo performance by reward as metric

| Dataset | Environment | BC | CQL | IQL | DT | TT | MOPO | MOReL | MBOP | Diffuser | AdaptDiffuser | Ours |
|---------|-------------|-----|-----|-----|-----|-----|------|-------|------|----------|---------------|------|
| Med-Expert | HalfCheetah | 55.2 | 91.6 | 86.7 | 86.8 | 94.0 | 63.3 | 53.3 | **105.9** | 88.9 | 89.6 | **98.7** |
| Med-Expert | Hopper | 52.5 | 105.4 | 91.5 | 107.6 | **110.0** | 23.7 | **108.7** | 55.1 | 103.3 | **111.6** | **111.2** |
| Med-Expert | Walker2d | **107.5** | 108.8 | 109.6 | 108.1 | 101.9 | 44.6 | 95.6 | 70.2 | **106.9** | 108.2 | 106.3 |
| Medium | HalfCheetah | 42.6 | 44.0 | **47.4** | 42.6 | 46.9 | 42.3 | 42.1 | 44.6 | 42.8 | 44.2 | 41.0 |
| Medium | Hopper | 52.9 | 58.5 | 66.3 | 67.6 | 61.1 | 28.0 | **95.4** | 48.8 | 74.3 | **96.6** | 83.8 |
| Medium | Walker2d | 75.3 | 72.5 | 78.3 | 74.0 | 79.0 | 17.8 | 77.8 | 41.0 | 79.6 | **84.4** | 80.6 |
| Med-Replay | HalfCheetah | 36.6 | **45.5** | 44.2 | 36.6 | 41.9 | 53.1 | 40.2 | 42.3 | 37.7 | 38.3 | 43.9 |
| Med-Replay | Hopper | 18.1 | 95.0 | 94.7 | 82.7 | 91.5 | 67.5 | **93.6** | 12.4 | **93.6** | 92.2 | **94.2** |
| Med-Replay | Walker2d | 26.0 | 77.2 | 73.9 | 66.6 | 82.6 | 39.0 | 49.8 | 9.7 | 70.6 | **84.7** | 85.3 |
| Average | | 51.9 | 77.6 | 77.0 | 74.7 | 78.9 | 42.1 | 72.9 | 47.8 | 77.5 | **83.40** | 82.89 |
| Average on Mixed Dataset | | 49.32 | 87.25 | 83.43 | 81.40 | 87.15 | 48.53 | 73.53 | 49.27 | 83.50 | 87.60 | **89.67** |

Table 1: **Performance on D4RL (Fu et al., 2020) MuJoCo environment using default reward metric (normalized average returns).** This table presents the the mean of 5 rollouts reward after learning from the D4RL dataset. Values over $95\%$ of the best are marked as **bold** of each dataset row. Baseline results are adopted from Liang et al. (2023).

We utilize MuJoCo tasks as a benchmark to assess the capability of our DOG when learning from diverse data with different quality levels sourced from the widely recognized D4RL datasets (Fu et al., 2020). To contextualize the efficacy of our technique, we compare it against a spectrum of contemporary algorithms encompassing various data-driven strategies. Overview of these methods are deferred to Sec. D. Comprehensive outcomes of this comparison can be found in Table 1.

Note that our scenario is more challenging compared to the offline RL baselines as we lack knowledge of the goal during the training phase. Our primary objective is to increase speed guide $\mathbf{grad} = \nabla_\Omega \mathrm{speed}(\Omega)$, aiming to generate a high-speed plan.

The findings are documented in Tab.1. In single dataset settings, our model demonstrates competitiveness with other offline RL algorithms. Specifically, on mixed datasets with expert data, our model consistently outperforms both Diffuser and AdaptDiffuser.

Our model excels particularly in mixed dataset scenarios. This substantiates our ability to sample from the distribution aligned with the goal while mitigating the impact of inferior samples. BC can be construed as the standard distribution, allowing us to surpass the original behavior. An enhancement in performance is observed when using Med-Replay in lieu of Medium, illustrating our proficiency in assimilating the high-value segments of diverse data without significant disturbance from the lower-value segments. This holds considerable significance for offline RL, especially in scenarios where acquiring high-quality expert data poses challenges.

## 5.4 Robotic movement control

Fig. 5 demonstrates that controlling robotic deployment involves calibrating properties like speed and height to comply with environmental constraints. Conventional RL techniques are typically limited to training with a single goal, and goal-conditioned RL primarily reaches specific states without extending control over the process. Here, we emphasize the aptitude of our method in altering the action sequence during the evaluation phase.

To illustrate, we deploy Hopper-v2, with the outcomes displayed in Fig. 5c - 5b, revealing various adaptations in speed and height upon the implementation of respective guidance. These modifications combine the accuracy of manual tuning with the merits of autonomous learning from demonstrations, all while staying within logical distribution ranges. There is no obligation to predefine the speed value, empowering the agent to make autonomous recalibrations to align with the target direction.

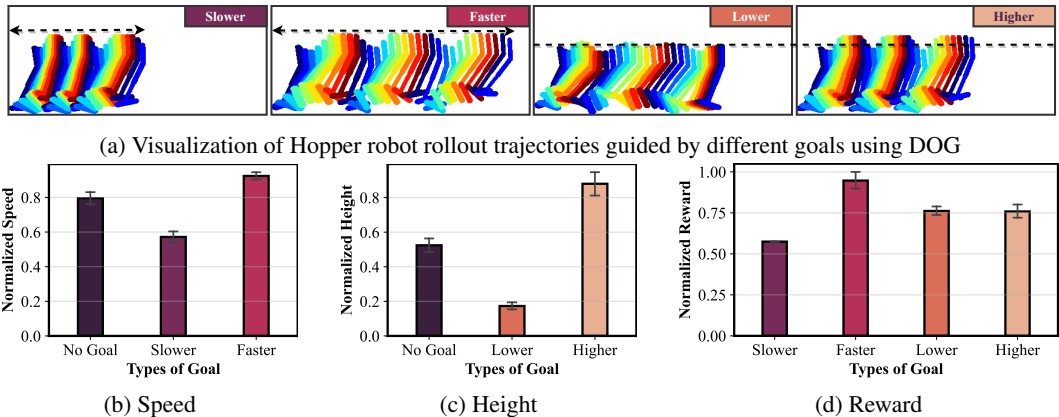

(a) Visualization of Hopper robot rollout trajectories guided by different goals using DOG

(b) Speed

(c) Height

(d) Reward

Figure 5: **Guidance of Robotic Movement.** (a) For the Hopper-v4 environment, we present guided rollouts depicted in 128 frames, displayed in intervals of 8 within a single figure. All four originate from the same state $s_0$, yet exhibit varied behavior under goal guidance. (b, c, d) The distribution of Speed, Height and original task reward under goal guidance.

## 5.5 ROBOT ARM CONTROL

We present result of interacting with the object with robot hand in this section. The goal of the original environment is to open a door with hand, which including sub steps: hold the handle and open as show in Fig. 6a. We use the "cloned" part of the D4RL dataset to train the planner model.

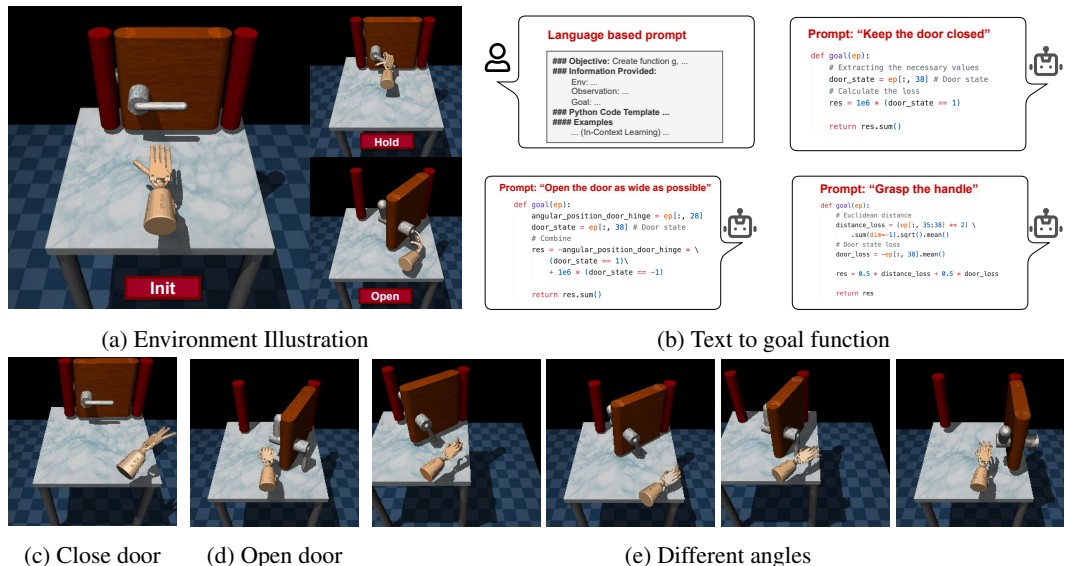

(a) Environment Illustration

(b) Text to goal function

(c) Close door    (d) Open door    (e) Different angles

Figure 6: **Demonstrations on D4RL Adroit Door-v1 environments**.

### 5.5.1 GOAL OF INTERACTING WITH OBJECT

In Fig. 6c-6e, we exhibit some example results of generating varied plans by assigning goals to different end states of the door. Notably, while the primary intention of this dataset is to accumulate data related to opening the door, it proves versatile enough to allow the formulation of plans that include keeping the door closed, opening it, and opening it to a specified angle.

### 5.5.2 FORMULATING GOAL FUNCTIONS WITH LLMS

Manually creating goal functions can be deemed a meticulous and laborious task. Here, we try to use a large language model (LLM) to translate abstract concepts into tangible goal functions. The observation space of the Android-Door-v1 environment harbors 39 distinct dimensions of information including angle, position, and speed, serving as an optimal candidate for testing the formulation of goal functions.

We employ GPT-4 32K (OpenAI, 2022) by feeding it prompts to generate requisite functions. The input prompt comprises our objective and comprehensive environmental data including each observation dimension's description and significance, coupled with several input-output pairs for in-context learning—a standard practice in prompt engineering (Min et al., 2021; 2022). Detailed input instances and dialogue history are available in Sec.G in the Appendix. Three prompts are input as depicted in Fig.6b. The outcomes confirm that the contemporary LLMs adeptly interpret our requirements and transfigure them into goal functions within our framework. And we only need to query once for one task.

## 6 RELATED WORK: DIFFUSION MODELS FOR DECISION MAKING

With the popularity of diffusion models in many generation tasks recently, there emerges a few attempts to leverage the power of diffusion models for decision-making problems. Wang et al. (2023); Pearce et al. (2023) used diffusion models to approximate more sophisticated policy distribution from offline RL datasets, achieving promising performance on GCRL tasks. By contrast, we use diffusion models to generate state trajectories. Janner et al. (2022) and Liang et al. (2023) proposed Diffuser and AdaptDiffuser, from which we have been inspired. While they focused on conventional offline RL/GCRL scheme, we extend to open-ended goal space by leveraging latest training-free guidance techniques (Yu et al., 2023b). Another key difference is that they used the diffusion model to generate both states and actions, thus need to struggle with the consistency between them (Liang et al., 2023). Besides, Ajay et al. (2022) and Dai et al. (2023) used diffusion models to generate state trajectories and use inverse kinematics to compute the action, similar to ours. However, they rely on classifier-free diffusion guidance (Ho & Salimans, 2021), meaning that the goal space need to be involved in training, thus it can not handle open-ended goals in evaluation.

## 7 CONCLUSIONS AND DISCUSSIONS

In this work, we proposed the scheme of open-ended goals for embodied AI and Diffusion for Open-ended Goals (DOG), a framework for open-ended task planning of embodied AI. Trained with general offline/demonstration data, DOG is featured with its capacity to handle novel goals that are not involved in training, thanks to the recent development of diffusion models and training-free guidance. While recent AI has been competitive with humans on natural language (Bubeck et al., 2023) and vision (Kirillov et al., 2023) tasks, currently embodied AI is far behind humans on e.g., cooking, room-cleaning and autonomous driving. Our work introduces a novel way of solving open-ended embodied tasks, which may shed light on cognitive neuroscientific research on understanding human intelligence (Taylor et al., 2016).

Our framework also comes with limitations. First, an energy function describing the goal (1) needs to be given. While we used human-defined function for the goals in this work, it could also be a neural network like a classifier (Dhariwal & Nichol, 2021). Future work should consider removing the method's dependence on humans. Another limitation is that DOG may fail to achieve a goal that cannot be accomplished by composing the existing state transitions in the training dataset. We argue that this is also highly challenging for a human, e.g., a normal person cannot make a helicopter, since an ability demands an accurate world model and extremely high computation power (Deisenroth & Rasmussen, 2011). Generalizability to a wider range of goals requires higher diversity in the offline dataset used for training. Nonetheless, we have demonstrated various training-free controllability such as speed and height regularization of moving robots using the D4RL dataset (Fu et al., 2020) which was not created for our purpose. Last but not least, future work should compromise multi-modalities including vision, audition, olfaction, tactile sensation, and text.

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

# A  MORE RELATED WORKS

## A.1  TASK OPEN-ENDEDNESS IN EMBODIED AI

The concept of task *open-endedness* (Ruiz-Mirazo et al., 2004; Taylor et al., 2016) (while the exact definition remains debatable) has long been recognized in biological learning and evolution. However, in machine learning and deep learning, models and algorithms until 2010's could only deal with single-objective tasks with real-world level difficulty (Krizhevsky et al., 2012; Silver et al., 2016). With the blossom with deep learning AI in recent few years, arguably represented by GPT-3 (Brown et al., 2020), AI researchers started to investigate the way to solve open-ended tasks by embodied AI in more realistic environments (Stooke et al., 2021; Bauer et al., 2023). One way is to exploit the planning capacity of large language models (LLMs) to decompose a task a into a sequence of primitive tasks described by nature language (e.g., "grab $\cdots$", "move to $\cdots$") (Brohan et al., 2023; Driess et al., 2023). However, the primitive tasks need to be pre-defined so that the robot can execute, thus limits its versatility. Moreover, nature languages may not be suitable for describing meticulous action movements such as shadow play using fingers. Another idea from Yu et al. (2023c) is to leverage the coding ability of LLMs to translate task instructions into reward functions, which can be used to compute low-level actions via model predictive control (Howell et al., 2022). However, this requires the LLM to know the specific robot used and it may struggle with high-dimensional action space. Besides relying on pre-trained LLMs, training large transformer (Vaswani et al., 2017) models from scratch for embodied decision-making on multi-tasks (or meta-learning) is also under investigation Stooke et al. (2021); Fan et al. (2022); Bauer et al. (2023). They differ from our method in terms of that their agents are heavily trained with a variety of human-defined goals (thus can generalize to novel goals) while our agents are trained without a goal. Finally, utilizing the power of diffusion models (Rombach et al., 2022; Dai et al., 2023) is another direction of exploration, which we defer to Sec. 6 for detailed discussion.

## A.2  MODEL-BASED PLANNING

Model-based planning methods searches (optimizes) a sequence of of state transitions that lead to desired outcome (e.g., more rewards) using a world model, typically by backpropagation (Deisenroth & Rasmussen, 2011; Parmas et al., 2018) or amortized inference (Hafner et al., 2019; Okada et al., 2020). The diffusion model in our framework serves for the same purpose, while employing totally distinct methodology. Model-based planning usually leverages state space models such as RNNs (Hafner et al., 2019) to model the state transition function in MDP explicitly, whereas our framework skips learning the exact state transition function, but straightforwardly generates state trajectories (with variant intervals) by using diffusion models and training-free guidance (Yu et al., 2023b). This is similar to people's planning in often cases, where one can imagine a trajectory of a basketball without an exact state-transition models or knowing Newton's law.

## A.3  GOAL-CONDITIONED DECISION-MAKING

A bunch of studies have worked on reinforcement learning (RL) and imitation learning (IL) in presence of a goal (Liu et al., 2022), known as goal-conditioned RL/IL (we refer to both as GCRL for simplicity). Studies of GCRL typically include controlling robots or mechanical arms to manipulate objects (Andrychowicz et al., 2017; Mendonca et al., 2021), going to a target position in a specific environment (Sharma et al., 2020; Yang et al., 2022). These work usually model the goal by a desired state to reach. Guided policy search (Levine & Koltun, 2013) and latent plans from play (Lynch et al., 2020; Rosete-Beas et al., 2023) share some high-level idea with ours, that is constraining the goal-directed policy close to that learned from offline experience so as to reduce search space. However, the key difference between GCRL and our work is that our training process does not involve a goal while can open-ended goals can be handled in evaluation, whereas GCRL pre-defines a goal space on which both training and evaluation are based, and treats the goal as input to the policy network. In short, unlike our framework, GCRL cannot handle goals undefined in training (e.g., keeping away from a state).

# B  IMPLEMENTATION DETAILS

## B.1  MODEL STRUCTURE

We follow Liang et al. (2023) and Janner et al. (2022) to use a a temporal U-Net (Ronneberger et al., 2015) with 6 repeated residual blocks is employed to model the noise of the diffusion process. Timestep embeddings are generated by a single fully-connected layer and added to the activation output after the first temporal convolution of each block.

## B.2  GOAL FUNCTION DESIGN SUMMARY

In this section, we provide a consolidated overview of the various goal functions implemented within our experiments, organized and summarized for coherent assimilation. The meticulous design of goal functions plays a pivotal role in shaping the trajectory of our experiments, facilitating nuanced interactions between the agent and its environment. It is essential to comprehend the multiplicity and specificity of these goal functions to appreciate their impact on the respective experimental outcomes. We aim to elucidate the intricate relations between observations, methods, and the corresponding goal function designs, aiding in a deeper understanding of the experimental setup and results.

To offer a structured perspective, we have tabulated all the goal functions, aligning them with their corresponding observations and methods, in Tab. 2. This table serves as a succinct reference, aiding readers in correlating the diversity of goal functions with the experimental setups and drawing insights from their interrelations.

# C  ENVIRONMENT INFORMATION

**Maze2D Envrionment (Fu et al., 2020)** serves as a navigation task platform where a 2D agent navigates from a randomly chosen start location to a predetermined goal location. Here, the agent is awarded a reward of 1 upon reaching the goal, with no intermediate rewards provided, necessitating efficient trajectory planning to secure optimal rewards.

This environment is structured to test the proficiency of offline RL algorithms in leveraging previously accumulated sub-trajectories to discern the optimal path to the designated goal. It offers three distinct maze layouts named "umaze," "medium," and "large." For the dataset we used in this paper, we build a maze with both height and width as 5 with no walls. Then generate 1 million steps followint the genration policy provided by Fu et al. (2020).

**MuJoCo Environment** developed by Todorov et al. (2012), is a physics engine esteemed for its ability to simulate intricate mechanical systems in real-time. It's primarily used for developing and testing algorithms in robotic and biomechanical contexts. Within MuJoCo, various tasks like Hopper, HalfCheetah, and Walker2d are available, each providing a unique set of challenges and learning opportunities for reinforcement learning models.

Each task within MuJoCo is associated with four different datasets: "medium," "random," "medium-replay," and "medium-expert" by D4RL dataset. The "medium" dataset is formulated by training a policy using a specific algorithm and accumulating 1M samples. The "random" dataset is derived from a policy initialized randomly.

The "medium-replay" dataset comprises all samples recorded during training until the policy attains a predefined performance level. Additionally, there exists a "medium-expert" dataset, representing a collection of more advanced, intricate scenarios and data points. These datasets are crucial for evaluating the adaptability, efficacy, and robustness of reinforcement learning algorithms under varying conditions and degrees of complexity.

**Adroit-Door Environment (Rajeswaran et al., 2017)** is designed as a sophisticated platform for interaction and manipulation tasks involving door objects. It presents an intricate set of challenges focused on analyzing the agent's capacity to interact with and manipulate objects effectively within defined parameters.

| Observations | Goal | Goal Function Design |
|---|---|---|
| Include observations $\{x, y, v_x, v_y\}$. | Goal as State | $(x - \hat{x})^2 + (y - \hat{y})^2$ |
| | Partial goal | $(x - \hat{x})^2$ |
| Include observations $\{x, y, v_x, v_y\}$. $(x, y)$ is the current position of the agent and $(v_x, v_y)$ is the velocity. | Distance - Go faraway | $-\sqrt{\sum_{i=1}^{N-1}(x_{i+1} - x_i)^2}$ |
| | Distance - Stay close | $\sqrt{\sum_{i=1}^{N-1}(x_{i+1} - x_i)^2}$ |
| | Non-specific Goal | $x$ |
| | Zero-shot transfer to new environment | $-\|x - x_0\| \cdot I(\|x - x_0\| < \sigma)$, where $\sigma = 1$ is the radius of points affected |
| | Hybrid goals | $\sum_i^{N_g} \eta_i \cdot g_i(\cdot)$ where $g_{1:N_g}$ is the series of goal functions and $\eta_i$ is the weight for each function. |
| Hopper HalfCheetah have different observations, while they both have dimension for the x-velocity and z-position of the robot tip. We denote the velocity as $v_x$ and z-position as $x_h$. | Faster | $-v_x$ (we consider one direction) |
| | Slower | $v_x$ |
| | Higher | $-x_h$ |
| | Lower | $x_h$ |
| Include observations $r$ as the angle of the hinge and a flag $o$ indicate whether the door is open, open as 1. | Open the door wide | $r$ |
| | Close the door | $o$ |
| | Open the door | $-o$ |

Table 2: Summary of Methods and Goal Function Design

This environment incorporates a complex observation space consisting of 39 dimensions, encompassing various aspects such as angle, position, and speed. This multi-dimensional space is conducive to examining the interaction dynamics between the agent and the door, providing insights into the efficiency and adaptability of the applied algorithms.

The environment aims to simulate real-world interactions with objects, making it a suitable platform to test reinforcement learning algorithms' efficacy in performing tasks with high precision and reliability. The tasks designed in this environment range from simple interactions, like opening and closing doors, to more complex ones requiring a fine-tuned understanding of the object's state and the surrounding environment.

## D  MUJOCO BASELINES

The Mujoco baselines include model-free RL algorithms like CQL Kumar et al. (2020) and IQL Kostrikov et al. (2021), return-modulated techniques such as the Decision Transformer (DT) Chen et al. (2021), and model-based RL algorithms like the Trajectory Transformer (TT) Janner et al. (2021), MOPO Yu et al. (2020), MOReL Kidambi et al. (2020), and MBOP Argenson & Dulac-Arnold (2020).

# E  ALGORITHM SUMMARY

We summarize the algorithm in Alg. 1.

---

**Algorithm 1** DOG

---

**Training Stage**

**Input:** Offline data $D = \{\tau_i\}_i^{N_d}$.
**Output:** Planner $p_\theta$ and Executor $p_\phi$
Learn unconditional Planner $p_\theta(\Omega \mid s_t)$ (Eq. 2)
Learn Executor $p_\phi(a_t \mid s_t, s')$ (Eq. 4) stage

---

**Deployment Stage**
**Input:** Goal description $g \in \mathcal{G}$.
**for** each step $t$ **do**
    (Optional) Accept new goal $g' \in \mathcal{G}$, set $g = g'$.
    **if** need to update plan **then**
        Calculate plan $\Omega^*$ with Eq. 3. Record plan step $t_p = t$.
    **end if**
    Decide next waypoint as $s' = \Omega^*[t - t_p]$.
    Select and execute action $a$ with executor $p_\phi(a_t \mid s_t, s')$.
**end for**

---

# F  LIST OF POSSIBLE EXECUTOR IMPLEMENTATIONS

1. **Hindsight Experience Replay (HER)-based (Andrychowicz et al., 2017).** The actor function, denoted by , takes as input the current state $s_t$ and an expected future state $s'$, producing an action $a_t$ as output. Training is finished by sampling state pair $\{s, s'\}$ within one episode with interval of $t \sim \text{Uniform}(1, ..., T_a)$, where $T_a$ is the max tolerance of state shifting. The recorded action of state $s$ would be used as predicting label.

2. **Goal-conditioned controller** Goal-conditioned RL study how to reach custom $s \in \mathcal{S}$, which is a great option for our underline controller since what we need is to achieve a waypoint $\hat{s}$ from current state $s_t$. Specifically, we adopt the Ma et al. (2022) as the underline controller for its great performance and easy implementation.

3. **Model Predictive Control (MPC) (Yu et al., 2023a)** We can learn an environmental model $E : \mathcal{S} \times \mathcal{A} \to \mathcal{S}$ so that next state $s_{k+1}$ can be estimated by $E(s_k, a_k)$. Then, we find the optimal $a_t$ as $\arg\min_{a_{\Delta t, t:t+\Delta t-1}} \ell(s_{t+\Delta t}, \hat{s})$.

# G  GPT PROMPT DEMONSTRATIONS

**LLM Prompt - Prefix**

```
### Objective:
Create a Python function, 'g', that returns the loss value, 'res',
    for a given input 'ep'. The function 'g' should be
    differentiable and minimizeable via gradient descent. This
    function is intended to construct a sequence \( \tau = \{s_1,
    ..., s_T\} \) such that \( \tau = \arg\min_\tau g(\tau) \) based
     on specified language-based requirements described in the 'Goal
    '.

### Information Provided:
- **Env: ** A text description of the environment
- **Observation ('s'):** Provides the shape and the meaning of each
     position in 'ep'.
- **Goal:** A text description outlining the objective for which 'g
    ' needs to be developed.

### Python Code Template:
def goal(ep):
        # code start
    res = ...
        # code end
    return res

#### Example 1
USER INPUT:
Env: Move a ball in 2d World
Observation: 4 dim
    0: position x of the ball
    1: position y
    2: velocity x
    3: velocity y
Goal: "Move the ball to (1, 3)"
YOUR REPLAY:
res = ((ep[-1,:2] - torch.tensor([1,3])) ** 2).sum()

#### Example 2
USER INPUT:
Env: Move a ball in a room
Observation: 4 dim
    0: position x
    1: position y
    2: velocity x
    3: velocity y
Goal: "Find the shortest path"
YOUR REPLAY:
res = ((ep[1:]-ep[:-1])[:2]**2).sum(dim=-1).sqrt().sum()

#### Example 3
USER INPUT:
Env: Driving a car in a 2D world
Observation: 2 dim
    0: velocity x of the car
    1: velocity y
Goal: "Move faster"
YOUR REPLAY:
res = ep[:,0].mean()

Now, Let Us Begin, my input:
```

**LLM Prompt - Example 1 (Open the door as wide as possible)**

```
Env:
        A 28 degree of freedom system which consists of a 24
           degrees of freedom ShadowHand and a 4 degree of freedom
           arm. The latch has significant dry friction and a bias
           torque that forces the door to stay closed.
Observation:
        0       Angular position of the vertical arm joint
        1       Angular position of the horizontal arm joint
        2       Roll angular value of the arm
        3       Angular position of the horizontal wrist joint
        4       Angular position of the vertical wrist joint
        5       Horizontal angular position of the MCP joint of the
           forefinger
        6       Vertical angular position of the MCP joint of the
           forefinge
        7       Angular position of the PIP joint of the forefinger
        8       Angular position of the DIP joint of the forefinger
        9       Horizontal angular position of the MCP joint of the
           middle finger
        10      Vertical angular position of the MCP joint of the
           middle finger
        11      Angular position of the PIP joint of the middle
           finger
        12      Angular position of the DIP joint of the middle
           finger
        ...
        32      x position of the handle of the door
        33      y position of the handle of the door
        34      z position of the handle of the door
        35      x positional difference from the palm of the hand
           to the door handle
        36      y positional difference from the palm of the hand
           to the door handle
        37      z positional difference from the palm of the hand
           to the door handle
        38      1 if the door is open, otherwise -1
Goal:
        Open the door as wide as possible
```

**LLM Prompt - Example 2 (Keep the door closed)**

```
Env:
        A 28 degree of freedom system which consists of a 24
           degrees of freedom ShadowHand and a 4 degree of freedom
           arm. The latch has significant dry friction and a bias
           torque that forces the door to stay closed.
Observation:
        0       Angular position of the vertical arm joint
        1       Angular position of the horizontal arm joint
        2       Roll angular value of the arm
        3       Angular position of the horizontal wrist joint
        ...
        38      1 if the door is open, otherwise -1
Goal:
        Keep the door closed
```

```
                    ┌─────────────────────────────────────────┐
                    │  LLM Prompt - Example 3 (Grasp the handle)│
          ┌─────────┴─────────────────────────────────────────┴──────────┐
          │                                                                │
          │  Env:                                                          │
          │          A 28 degree of freedom system which consists of a 24  │
          │             degrees of freedom ShadowHand and a 4 degree of    │
          │             freedom arm. The latch has significant dry         │
          │             friction and a bias torque that forces the door    │
          │             to stay closed.                                    │
          │  Observation:                                                  │
          │          0        Angular position of the vertical arm joint   │
          │          1        Angular position of the horizontal arm joint │
          │          2        Roll angular value of the arm                │
          │          3        Angular position of the horizontal wrist joint│
          │          ...                                                   │
          │          38       1 if the door is open, otherwise -1          │
          │  Goal:                                                         │
          │          Grasp the handle                                      │
          │                                                                │
          └────────────────────────────────────────────────────────────────┘
```

