# OpenReview forum: "Toward Open-ended Embodied Tasks Solving"
_ICLR.cc/2024/Conference — Submitted to ICLR 2024_

### Official Review · Reviewer_kQcF · 2023-10-21

**Soundness:** 1 poor
**Presentation:** 1 poor
**Contribution:** 2 fair
**Rating:** 3
**Confidence:** 2

**Summary:**

The authors propose Diffusion for Open-ended Goals (DOG), a framework for learning various goal-related task capabilities using the diffusion models approach. In particular, they call upon "training-free" guidance capabilities to perform planning and control. They show tha capabilities of their framework in various setups, from goal-reaching in 2D space to Hopper and robotic door manipulation, even from language-based prompts.

**Strengths:**

Using diffusion models to perform Goal-Conditioned Reinforcement Learning (GCRL) looks like a good idea, although the authors are not the first to put it forward, and are not clear about the novelty they bring.

**Weaknesses:**

- The paper is very unclear about the "training-free" property of the framework, which seems to be the main contribution with respect to Janner et al. (2022) and Liang et al. (2023).

- The paper is unclear about the difference between defining an energy function by hand and defining a goal-related reward function by hand.

- The experiments about using language-based prompts are not explained at all

- The paper is written poorly and often in a non-scientific way with a lot of unnecessary over-statements (see below).

**Questions:**

- could the authors explain the difference between defining energy functions for partial goals, negative goals, sequence goals, hybrid goals as they do in the bottom of page 3 and defining goal-related reward functions that do the same, as often done in GCRL? If they argue that this makes a difference, can they backup this claim with comparative experiments?

- "This way of defining a goal is fundamentally different from the conventional way of treating the goal as a variable" -> could the authors explain why it is "fundamentally different"? You could have a goal represented with "Clean = True", this would be using a boolean variable, or "clean level = X", that would be using a scalar variable. Doesn't your energy function (c or g?) compute something similar?

- p6: the list of examples should be reordered to match the order of the figure (or vice versa). Besides, the corresponding energy functions should be given. The examples themselves need to be better explained.

For instance "our objective isn’t state-specific but aims at maximizing or minimizing some certain property," this is a completely unspecific sentence, and an over-statement. Can your system do whatever you ask it to do? What are the limits?

- in the related work, the authors connect training-free guidance to (Yu et al., 2023b): "and training-free guidance (Yu et al., 2023b)" But the paragraph where they speak of training-free guidance (p2) does not even mention (Yu et al., 2023b). This point must be explained more clearly, as it seems to be the central contribution of the paper.

-could the authors extend the related work and better explain the difference betwen their work and Janner et al. (2022) and Liang et al. (2023)? Putting it before the experimental study would help, in my opinion.

- Table 1: why is the authors' method performing worse than competitors in the medium dataset?

- could the authors explain how they transformed language-based prompts into a goal-related energy function?


- Appendix F, HER is not an executor function, it is a method to leverage failed trajectories in GCRL. I don't understand the authors' description at all, we clearly do not see HER the same way...


## Over-statements:

- p2: "This work is a trial to approach the ultimate embodied AI that can assist people with diverse tasks such as healthcare, driving, and housework, though further endeavor is, of course, still largely demanded." So, is this work a successful trial? Clearly, it does not approach this ultimate embodied AI it describes. This is a huge over-statement, and the authors should describe in the introduction the goals they truly manage to achieve.

- p5: "By doing so, it expedites the sampling process and ensures adherence to defined objectives, proficiently fulfilling our requirements and presenting a viable resolution to our challenges" -> only politicians write this way! :) Be concrete about the positive sides of your method, and do not be afraid of listing limitations, you are doing science, not selling something

- p6: "Fig. 4h exemplifies our algorithm’s prowess" -> sorry, I don't consider this as a prowess.

- p7: This holds considerable significance for offline RL: what makes you believe this is so? Without properly backing up this claim with results, I see this as an over-statement.

- "Our work introduces a novel way of solving open-ended embodied tasks, which may shed light on cognitive neuroscientific research on understanding human intelligence (Taylor et al., 2016)." -> This is a huge over-statement. Can the authors explain in which way they believe their work can concretely shed light on cognitive neuroscientific research on understanding human intelligence?


## Local issues and typos ##

- p2: These works are worthy of recognition on the path to embodied AI that can truly tackle open-ended tasks.
Do the authors use "task" and "goal" interchangeably?

- p2: from somewhere.). -> from somewhere).

- Bayesian formula -> Bayes formula

- p3, in (1) the energy function is written g, but in the examples below it is written c


- "Generative models such as diffusion models for image generation can “draw” the pixels with any color, whereas a real-world painter can only use the colors available in the palette". -> This makes no sense. How can a generative model draw an image with pixels outside the computer's palette?

- "This is different from traditional machine learning tasks, where the focus is on finding a single optimal output for a given input." -> By traditional, do you mean supervised learning? Reinforcement learning is traditional (rooted in Bellman, 1957) but it corresponds to what you describe in the previous sentence.

- "Oftentimes we care about multiple steps rather than a single state, e.g, asking a robot to move in circles. Therefore, tackling
open-ended task goals in embodied AI becomes challenging." -> this is very poorly written. Ask yourself what is your point and write what your point is, avoid being only allusive.

- p3: dimension (Here -> here

- discrete actions.) -> actions).

- you don't need a "nanny" robot to clean or tidy a house, a anny is to take care about children.

- p4: the task is achieve -> is to achieve

- p5: concatate -> concatenate

- p5: plan executing -> execution

- p5: Several optional can be made for the plan executor -> there are several options...
By the way, these options should be described in the main text

- p5: Since our executor to multiple-step actions, -> something is missing

- p5: to the Appendix: which one?

- p5: the generating policy consistent with D4RL: what do you mean? I don't understand.

- p6: we present 100 episodes both from the same start point (marked as green star). -> why "both"? It makes no sense.

- p7: the the mean

- p8: which including sub steps: -> includes

- Appendix B, try to fit Table 2 into Appendix B.2

- Appendix C, about Mujoco environment: "using a specific algorithm and acculumating 1M samples" -> which  specific algorithm do you mean?

- Appendix F:

"by , takes" -> something is missing

we adopt the (Ma et al. (2022) as ..." -> something is missing

---

### Official Review · Reviewer_z7mQ · 2023-10-30

**Soundness:** 3 good
**Presentation:** 3 good
**Contribution:** 3 good
**Rating:** 5
**Confidence:** 4

**Summary:**

This paper introduces a new framework called Diffusion for Open-Ended Goals (DOG) that enables embodied AI to plan and act flexibly and dynamically for open-ended task goals. The authors argue that current approaches are limited by the conventional goal-as-an-input paradigm, which restricts the flexibility of goals. DOG synergizes the generative prowess of diffusion models with state-of-the-art, training-free guidance techniques to adaptively perform online planning and control. The framework is trained with general offline/demonstration data and is featured with its capacity to handle novel goals that are not involved in training. An energy function describing the goal needs to be given.

**Strengths:**

1. The paper presents very interesting evaluation results that demonstrate the effectiveness of DOG in enhancing embodied AI's adaptability and competency in tackling open-ended goals.

2. The authors also discuss the limitations of DOG and suggest future research directions. Overall, this paper contributes to the advancement of embodied AI and has the potential to inspire further research in this area.

**Weaknesses:**

Please refer to Questions section.

**Questions:**

1. What are the limitations of the DOG framework when applied to more complex tasks (like multimodal control tasks), and how might they be addressed in future research?

2. Can diffusion model used in your work be replaced with any another model? What is the advantage of the diffusion model in your setting?

3. Does DOG suitable for real world embodied AI problems? What are the challenges? Can you provide some insights on applying DOG to the real world?

---

### Official Review · Reviewer_Q6JW · 2023-10-30

**Soundness:** 3 good
**Presentation:** 3 good
**Contribution:** 3 good
**Rating:** 6
**Confidence:** 3

**Summary:**

In this work, the authors are concerned with learning open-ended task planning. The authors introduce Diffusion for Open-ended Goals (DOG), a method that is capable of solving this problem through the use of diffusion models and training-free guidance. The authors conduct qualitative experiments on DOG, testing it in various tasks where an agent must calculate a trajectory to achieve a novel goal in an open-world and door-opening environment. Quantitative results are also given in MuJoCo experiments that compare DOG to previous reinforcement learning and diffusion model baselines.

**Strengths:**

- The mechanism the authors made to swap out the goal at test time is pretty interesting.
- The author's experiments support their motivation.
- The figures in the experiment section are very well done and do a good job of illustrating the capabilities of DOG.
- Section 5.5.2 is nicely placed with contemporary work.

**Weaknesses:**

My major comments and questions are as follows.

"Mixed Dataset" is not defined. Without this Table 1 has a lot less impact as under the Mixed Dataset DOG outperforms previous work.

Some details are missing.
- The authors note "$s_\theta(x^n,t)$ ...", but $x^n$ is not defined.
- Why is $p(c|\Omega^n) \propto exp(-l(x^n))$? Please cite a reference or clarify.
- Can the authors add more details to what they mean within the norm of equation 2? More specifically how does one get a loss from $\Omega$ as it is a sequence of observations?
- In the conditional score function during the test phase, what is $\eta$?
- $\bar{\Omega}^0$ is defined but never used.
- What is "speed guide" or the function "speed"?
- 'We use the “cloned” part of the D4RL dataset' - What does cloned mean here?

**Questions:**

The major comments and questions are given in the weaknesses section. Beyond those, minor comments and questions are as follows.

Small errors
- $\Omega^n$ is first defined as "noisy data" and then "random noise" immediately after.
- Table 1's caption has "the the mean".

Question
- In 5.2, what would happen if the obstacles were enabled?

Suggestions (up to authors if they want to incorporate) to improve readability
- In Figure 1 instead of "Our Framework" use "DOG"
- Figures should be self-contained and include captions.
- In the caption of Figure 2, "goal-conditioned decision-making" is said to be on the left, but it would be better if it said "top-left".

---

### Official Review · Reviewer_HKQT · 2023-11-01

**Soundness:** 2 fair
**Presentation:** 1 poor
**Contribution:** 1 poor
**Rating:** 3
**Confidence:** 4

**Summary:**

This paper proposes using diffusion models to generate a set of trajectories given a goal specification. The paper starts by discussing about the filed in the embodied AI, followed by illustrating 4 types of different goal specifications. Finally, the paper concludes by running experiments to show the method performs well.

**Strengths:**

1. Overall the paper proposes an approach that follows the line of research direction of using diffusion models to generate feasible trajectories for the embodied AI tasks.

**Weaknesses:**

1. The paper writing is poor. For example, in the introduction section, it talks about something big in the sense that it wants to solve embodied AI tasks, then suddenly switches to something that uses diffusion models to generate the trajectories. I think the introduction section should be rewritten.
2. The proposed method is incremental compared to Janner at el. or AdaptDiffuser. Based on table 1, we can see that AdaptDiffuser and the proposed method have the same performance. In addition, it is unclear to me how the proposed model can combine GPT to process the natural language instruction to generate the trajectory (in Figure 6). As a result, the method is not convincing.
3. The paper says that it wants to solve "open-ended" problem, but in reality, the method is constrained to solve a fixed set of the problem, and there is no notion of unseen environment, making it hard to evaluate that if the proposed method can truly solve the open-ended questions.
4. Overall, I think this paper does not meet the bar of ICLR, and should be polished in the next round of reviewing.

**Questions:**

Please address the above comments.

---

### Author Response · Authors · 2023-11-23

We deeply appreciate your comments and suggestions. We will take more time and efforts to improve the research accordingly.

---

### Meta-Review · Area_Chair_hGR2 · 2023-12-05

**Metareview:**

This work touches on an exciting area of research, namely using generative models to discover behaviors not present in an offline dataset and then generalizing to new tasks. Unfortunately the reviewers had many questions around clarity which were not addressed during the rebuttal phase, and the experiments do not sufficiently demonstrate the generalization capabilities or open-endedness of the approach. If these areas are improved the work could be accepted to a future ICLR-like conference.

**Justification For Why Not Higher Score:**

Clarity was the main issue, reviewers were unable to assess novelty as they could not understand how different components combined. For instance, the results look very similar to a baseline (AdaptDiffuser) and the authors claim they also use GPT4 but reviewers could not see how this was possible in DOG but not the baselines. In addition the experiments were not convincing with small improvements on fairly out of date tasks that did not fully demonstrate the capabilities of this diffusion based approach.

**Justification For Why Not Lower Score:**

N/A

---

### Decision · Program_Chairs · 2024-01-16

Reject